# Real World Patterns of Antimicrobial Use and Microbiology Investigations in Patients with Sepsis outside the Critical Care Unit: Secondary Analysis of Three Nation-Wide Point Prevalence Studies

**DOI:** 10.3390/jcm8091337

**Published:** 2019-08-29

**Authors:** Maja Kopczynska, Ben Sharif, Harry Unwin, John Lynch, Andrew Forrester, Claudia Zeicu, Sian Cleaver, Svetlana Kulikouskaya, Tom Chandy, Eshen Ang, Emily Murphy, Umair Asim, Bethany Payne, Jessica Nicholas, Alessia Waller, Aimee Owen, Zhao Xuan Tan, Robert Ross, Jack Wellington, Yahya Amjad, Vidhi Unadkat, Faris Hussain, Jessica Smith, Sashiananthan Ganesananthan, Harriet Penney, Joy Inns, Carys Gilbert, Nicholas Doyle, Amit Kurani, Thomas Grother, Paul McNulty, Angelica Sharma, Tamas Szakmany

**Affiliations:** 1Department of Anaesthesia, Intensive Care and Pain Medicine, Division of Population Medicine, Cardiff University, Heath Park Campus, Cardiff CF14 4XN, UK; 2Anaesthetic Directorate, Aneurin Bevan University Health Board, Royal Gwent Hospital, Cardiff Road, Newport, Gwent NP20 2UB, UK

**Keywords:** sepsis, antibiotics, microbiology, investigations

## Abstract

Recent description of the microbiology of sepsis on the wards or information on the real-life antibiotic choices used in sepsis is lacking. There is growing concern of the indiscriminate use of antibiotics and omission of microbiological investigations in the management of septic patients. We performed a secondary analysis of three annual 24-h point-prevalence studies on the general wards across all Welsh acute hospitals in years 2016–2018. Data were collected on patient demographics, as well as radiological, laboratory and microbiological data within 48-h of the study. We screened 19,453 patients over the three 24 h study periods and recruited 1252 patients who fulfilled the entry criteria. 775 (64.9%) patients were treated with intravenous antibiotics. Only in 33.65% (421/1252) of all recruited patients did healthcare providers obtain blood cultures; in 25.64% (321/1252) urine cultures; in 8.63% (108/1252) sputum cultures; in 6.79% (85/1252) wound cultures; in 15.25% (191/1252) other cultures. Out of the recruited patients, 59.1% (740/1252) fulfilled SEPSIS-3 criteria. Patients with SEPSIS-3 criteria were significantly more likely to receive antibiotics than the non-septic cohort (*p* < 0.0001). In a multivariable regression analysis increase in SOFA score, increased number of SIRS criteria and the use of the official sepsis screening tool were associated with antibiotic administration, however obtaining microbiology cultures was not. Our study shows that antibiotics prescription practice is not accompanied by microbiological investigations. A significant proportion of sepsis patients are still at risk of not receiving appropriate antibiotics treatment and microbiological investigations; this may be improved by a more thorough implementation of sepsis screening tools.

## 1. Introduction

Sepsis, defined as dysregulated host response leading to life threatening organ dysfunction secondary to infection, has been thought to contribute significantly to in-hospital mortality [1,2]. Recent data suggests that despite increased awareness of this condition, the incidence of sepsis in the last decade has remained high and its associated morbidity and mortality has increased. This is considered to be a consequence of ageing populations which are changing cohort characteristics to favour the admission of patients with lower physiological reserve [3].

The early administration of resuscitation bundles, including appropriate antibiotics, has been demonstrated to improve patient outcomes [4]. This has led to several national and international quality improvement initiatives. Such initiatives have been established to facilitate early disease recognition and rapid initiation of treatment [5,6]. In response to sepsis in England and Wales, the NHS has adopted a standardised sepsis screening tool and Sepsis-Six bundle [7]. We have investigated the effectiveness of this approach previously in our series of point prevalence studies [8,9,10].

Despite the evidence of the potential benefit of these initiatives, there is growing concern surrounding the indiscriminate use of antibiotics and omission of microbiological investigations in patient management [11,12,13]. Additionally, there is no recent description of the microbiology of sepsis on the wards or information on the real-life antibiotic choices used in sepsis.

In light of these controversies, the primary objective of the study was to investigate antibiotic prescribing practices on general wards and emergency departments (ED) in at-risk population of patients in acute hospitals in Wales. The secondary objective was analysis of outcomes of microbiological investigations of sepsis patients using our comprehensive database.

## 2. Experimental Section

### 2.1. Study Design and Participants

Secondary analysis of patient episodes was performed on the patient population recruited into three annual 24-h point-prevalence studies on the general wards and ED across all Welsh acute hospitals in the years 2016, 2017 and 2018. In order to be entered into the study, each hospital was required to have a 24/7 consultant supervised ED and the ability to admit and treat any acutely unwell patient. We recruited patients with National Early Warning Score (NEWS) ≥3 and proven or suspected infection documented in the clinical notes. Those who were under 18, in mental health or critical care units were excluded as these patients are not covered by the NEWS system in the Welsh hospitals.

Data were collected using a digital platform, the methodology and performance of which is described in detail in our previous studies [8,14]. The data were collected from medical and nursing records and comprised of patient demographics, baseline pre-admission characteristics, clinical observations, as well as radiological, laboratory and microbiological data within 48-h of the study. The completion of sepsis screening tools, sepsis care bundles and antibiotic treatment were also recorded. Follow-up data collection was continued up until 90 days post-study.

Ethical approval was given by the South Wales Regional Ethics Committee (16/WA/0071). Written informed consent was gained from each patient, or by proxy for those who lacked capacity. The trial was registered prospectively at www.isrctn.com (ISRCTN86502304). 

### 2.2. Statistical Analysis

Categorical variables are described as proportions and are compared using the Chi-square test. Continuous variables are described as median and inter-quartile range. A two-tailed *p*-value < 0.05 was considered statistically significant. To identify factors associated with antibiotic administration we performed a multivariable logistic regression analysis with backwards elimination method, using antibiotic administration as a dependent variable. The likelihood ratio test was used in the backward elimination method using a significance level of *p*-value < 0.05. We only considered the main effects in this analysis; interaction terms were not included in the model. We determined the goodness-of-fit of the model using the Hosmer–Lemeshow test. The results of the multivariable analysis are shown as odds ratios (OR) and 95% confidence interval (95% CI). All statistical tests were calculated using SPSS 25.0 (SPSS Inc., Chicago, IL, USA).

## 3. Results

### 3.1. Patient Characteristics

We screened 19,453 patients over three 24-h point-prevalence study periods in 14 acute hospitals in Wales in 2016, 2017 and 2018. We recruited 1252 patients who fulfilled the study criteria. Patient characteristics are summarised in Table 1, with a more detailed comparison of the patients recruited in the ED and the ward provided in Appendix A. We had information about the use of antibiotics for 1195 patients.

### 3.2. Antibiotics Characteristics

Out of 1,195 patients, 775 (64.9%) were treated with intravenous (IV) antibiotics. Majority of patients were treated with one IV antibiotic (median 1, range 0–5). The antibiotics used were broad spectrum, with Piperacillin/Tazobactam (Tazocin) being the most commonly prescribed (40.5% of patients treated with antibiotics, 314/775). The most frequently prescribed antibiotics are presented in Figure 1. Tazocin was more commonly used as a monotherapy, however, in a significant proportion of patients, it was used in addition to other antibiotics (32.1%, 101/314). Tazocin was most commonly used in combination with Metronidazole and Clarithromycin (19/101 and 16/101, respectively).

### 3.3. Organism Characteristics

The vast majority of patients did not undergo microbiological investigations. Only in 33.65% (421/1252) of all recruited patients did healthcare providers obtain blood cultures; in 25.64% (321/1252) urine cultures; in 8.63% (108/1252) sputum cultures; in 6.79% (85/1252) wound cultures; and in 15.25% (191/1252) other cultures. The yield of the cultures was very low with positive result in only 13.06% (55/421) blood cultures, 13.71% (44/321) urine cultures, 45.37% (49/108) sputum cultures, 32.94% (28/85) wound cultures and 8.90% (17/191) other cultures. The most commonly identified organism in blood cultures was *E. coli* (17 out of 421 cultures) and Gram-positive cocci (16 out of 421 cultures), the latter most commonly interpreted as contamination. None of the organisms were characterized as multi-resistant. Detailed microbiological results are presented in Table 2.

### 3.4. Antibiotics Use in Patients Fulfilling SEPSIS-3 Criteria

Out of the recruited patients, 59.1% (740/1252) fulfilled SEPSIS-3 criteria of Sequential Organ Failure Assessment (SOFA) score of two or above. Patients with SEPSIS-3 criteria were significantly more likely to receive antibiotics than the non-septic cohort (490/740 of septic patients received antibiotics in comparison to 285/512 non-septic patients *p* < 0.0001) (Figure 2). Nevertheless, 33.78% (250/740) of patients fulfilling SEPSIS-3 criteria did not receive any antibiotics. The vast majority of this cohort of patients had no microbiological investigations done. Only 24.8% (62/250) had blood cultures, 10% (25/250) sputum cultures, 23.2% (58/250) urine cultures, 6.8% (17/250) wound cultures and 14.4% (36/250) other cultures. These results are similar for other sepsis screening tools, such as Systemic Inflammatory Response Syndrome (SIRS), quick SOFA (qSOFA) and Red Flag Sepsis (data not shown).

### 3.5. Antibiotics and Microbiological Investigations

Analysing only patients who received antibiotics, we found that 49.75% (300/603) had blood cultures, 13.81% (70/507) had sputum cultures, 35.47% (210/592) had urine cultures, 9.01% (53/588) had wound cultures and 13.42% (104/775) had other microbiological investigations. We also found that patients who received antibiotics were more likely to have obtained blood and urine cultures in comparison to the patients with no antibiotic treatment (*p* < 0.0001 and *p* = 0.04, respectively). No significant difference in sputum, wound or other microbiological investigations was noticed for this cohort. On the other hand, a significant proportion of patient who underwent investigations did not receive antibiotics. We found that 28.74% (121/421) of patients with blood, 35.19% (38/108) sputum, 34.65% (111/321) urine and 40.57% (71/175) other cultures were not prescribed any antibiotics. Further details on how the presumed site of infection influenced antibiotic therapy and microbiology investigations is provided in Appendix A.

We also investigated whether the use of official screening tools had an impact on antibiotic prescription practice. We found that the screening tools have been completed for only 17.7% (221/1252) of patients, despite high NEWS score and documented suspicion of infection. Interestingly, the screened patients were significantly more likely to receive antibiotics in comparison to non-screened patients (*p* < 0.0001) and to have blood culture (*p* = 0.001) but no other microbiological investigations.

### 3.6. Factors Influencing Antibiotics Administration

We used a binary logistic regression model to independently assess variables that in a univariate analysis were associated with antibiotic use and we felt were clinically important. Consequently, we included frailty score, SOFA score, number of SIRS criteria present, any cultures obtained and official screening tool completed. Hosmer–Lemeshow test indicated a good model fit. Increase in SOFA score (OR 1.078; 95% CI 1.008–1.152), increased number of SIRS criteria present (OR 1.225; 95% CI 1.080–1.389) and completion of the official screening tool (OR 2.949; 95% CI 1.996–4.358) were independently associated with antibiotic administration, however increased frailty score or microbiological sample collection was not.

## 4. Discussion

Our study shows that antibiotic prescription practice for patients with suspected sepsis is not accompanied by microbiological investigations. The majority of recruited patients with suspicion or confirmed infection, did not undergo microbiological investigations and were prescribed broad spectrum antibiotics.

Our study highlights the current practices of antibiotics prescribing in an era when awareness of sepsis is raised and in a healthcare system which has adopted a nation-wide screening tool and escalation process for the condition [7]. Patient characteristics, demographics and co-morbidities were similar to recently published large scale retrospective analyses and multi-national studies of similar methodology [4,15,16]. Antibiotic administration was common in this at-risk population, similarly to the IMPRESS study, however other elements of the Sepsis-Six bundle or the 3-h sepsis bundle appeared to be lacking [4,5,17]. In line with previous results of large quality improvement initiatives, the use of a checklist-based screening tool was associated with better adherence to microbiology sampling and antibiotic administration guidance [6].

The implementation of the sepsis bundles both within the UK and internationally have highlighted a potential drawback of the “treat first, ask questions later” approach, which fails to take into account the several factors which could influence the bedside clinicians, resulting in less than ideal practice [4,6]. First, the diagnosis of sepsis is notoriously difficult with limited clinical and laboratory information available. In our previous work using the same patient population, we have shown that the different clinical screening criteria commonly used outside of the critical care units capture different patient populations and that combining the SOFA- and SIRS-based approaches would identify around 80% of the patients at high risk of death [10]. Our findings that both SIRS and SOFA scores were independently associated with antibiotic administration are encouraging and could highlight a potential way forward for everyday clinical practice. Second, antibiotic use can have significant adverse consequences, from anaphylaxis, as highlighted recently by the NAP6 findings in the UK, to unnecessary exposure to antibiotics if a patient does not have a bacterial infection, or the treatment is continued beyond the clinically indicated duration [18,19]. Our findings, that broad spectrum antibiotic use was common without appropriate microbiological investigations is alarming, especially as it is corroborated by international data [4,20]. It has been shown, that patients with septic shock would benefit from early administration of broad-spectrum antibiotics, however convincing data on the less severe sepsis population is lacking [6,15,17]. In a recent randomised controlled trial of community-acquired sepsis with very similar patient population characteristics to our study, early (within 1-h of recognition) broad spectrum antibiotic administration failed to improve clinical outcomes compared to the usual care group which received antibiotics after assessment in the ED [21]. We have shown, that severe infection leading to organ dysfunction per se is relatively rarely attributable to death in the same patient population [22]. This finding was echoed in a recent study from the US, where Rhee et al. found that although sepsis was present in 52% of terminal hospitalisations, underlying causes of death were related to severe chronic comorbidities and most sepsis-associated deaths were unlikely to be preventable through better hospital-based care [23].

Early identification of the infectious organism could allow more targeted and effective treatment. This could also facilitate the use of antibiotics with narrower spectra or early antibiotics switching instead of using broad-spectrum antibiotics, an approach advocated in sepsis for over a decade [24,25]. Clinically, the nature of the organism triggering sepsis appears to have considerable prognostic significance on the ICU [26,27,28]. It can have an impact on clinical presentation as well as sepsis morbidity and mortality. Moreover, the nature of the infection can determine the mechanism of host response both to the pathogen and initiated therapy [29]. Targeted antimicrobial therapy would reduce the risk of development of multidrug-resistant organisms as well as antibiotics-related infections such as *Clostridium difficile* diarrhoea [30]. In a previous report, we found that over 98% of organisms responsible for positive blood cultures are sensitive to the combination of Tazocin and Gentamicin [31]. This might explain the preferential use of Tazocin in our current sample. However, in the same report it has been emphasised that before antibiotic administration, appropriate microbiology cultures should be obtained, which has not happened in the majority of our cases [31].

The lack of microbiological investigations could be influenced by the low yield of detection of organisms of the gold standard” microbiological investigations. In our study, the yield of blood cultures, which are a part of Sepsis Six bundle, was only 13.06%, similar to other reports [15,30,32]. This confirms the need to develop more sensitive diagnostic microbiological investigations. Emerging technologies such as next-generation sequencing, PCR-Electrospray Ionization mass spectrometry or use of various ‘-omics’ techniques [30] might change the approach to current microbiology standards and improve the detection of organisms in the future.

The strengths of our study include the participation of centres all across Wales, including both academic and general district hospitals, as well as using prospective data collection methods, resulting in a clinically rich and complete dataset.

Our study has some limitations. Firstly, patients with sepsis who had NEWS below 3 could have been missed and not recruited into our study [33,34]. However, recent data suggest that outside of the intensive care setting, the NEWS cut-off of 3 may be the most optimal trigger for sepsis screening [35]. Secondly, due to the 24-h point-prevalence character of our study we were not able to obtain information on how long the broad-spectrum antibiotics we used for in-patients’ treatment and whether an optimal step-down management was implemented. Lastly, due to the prospective nature of the study, the data collector could have possibly been recruiting patients before their team undertook investigations and management steps, especially during the night shift. However, we collected the microbiology results retrospectively and also reviewed the prescription charts where they were available to minimise this bias.

## 5. Conclusions

In conclusion, a significant proportion of sepsis patients are still at risk of not receiving appropriate antibiotics treatment and microbiological investigations. This could be improved by a more thorough implementation of sepsis screening tools. In addition, adopting both SIRS and SOFA clinical criteria may help to identify the high-risk population, where microbiological investigations should accompany appropriate antimicrobial therapy.

## Figures and Tables

**Figure 1 jcm-08-01337-f001:**
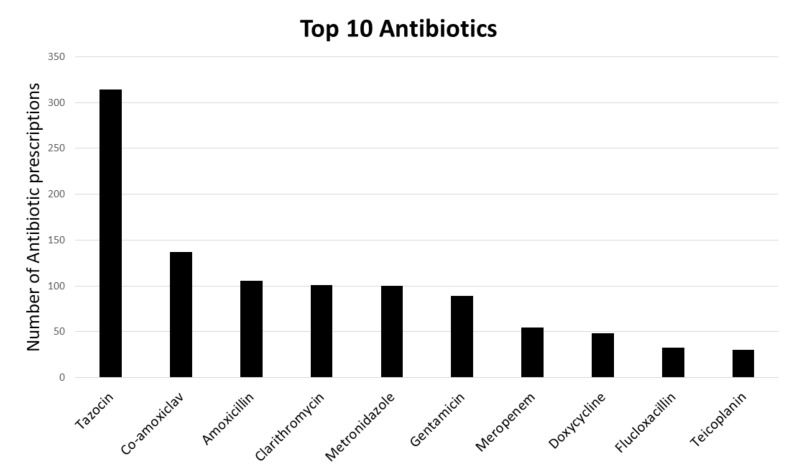
The most frequently prescribed antibiotics. Bars represent the number of times each antibiotic was prescribed.

**Figure 2 jcm-08-01337-f002:**
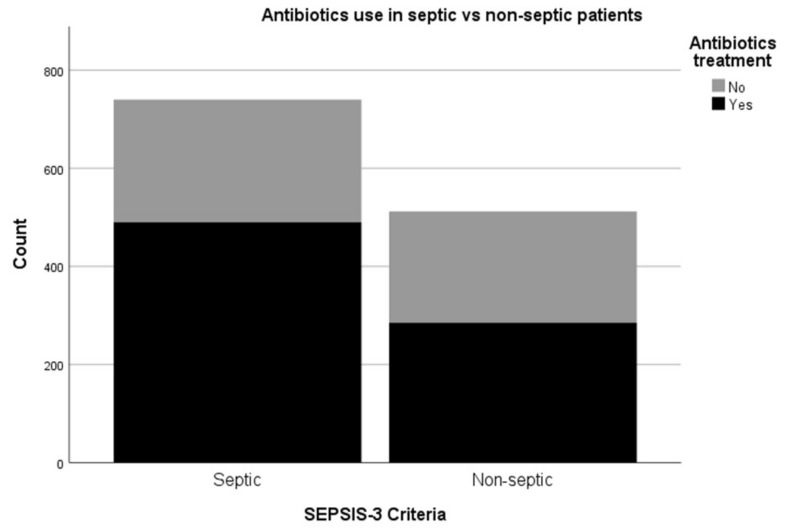
Comparison between antibiotics use between patients fulfilling SEPSIS-3 criteria (septic) and patients not fulfilling SEPSIS-3 criteria (non-septic). Septic patients were significantly more likely to receive antibiotics than the non-septic cohort *p* < 0.0001.

**Table 1 jcm-08-01337-t001:** Patient demographics information.

	Median	Interquartile Range (Q3–Q1)
Age (years)	73	22
Number of co-morbidities (n)	1	2
Frailty Score	5	3
NEWS score	4	3
SIRS score	2	2
SOFA score	2	2
	Number of Patients, n (%)	
Sex, male	624 (49.8)	
SEPSIS-3 criteria present	740 (59.1)	
ED admission	193 (15.4)	
General medical admission	544 (43.5)	
General surgical admission	197 (15.7)	
Admission to other ward	321 (25.6)	
Ceiling of care documented	263 (21.0)	
DNA-CPR documented	305 (24.4)	
COPD	347 (27.7)	
Diabetes	262 (20.9)	
Heart failure	144 (11.5)	
Hypertension	417 (33.3)	
Ischaemic heart disease	210 (16.8)	
Liver disease	43 (3.4)	
Neuromuscular disease	40 (3.2)	
Recent chemotherapy	50 (4.0)	
Chronic antibiotics	94 (7.5)	

NEWS, National Early Warning Score; SIRS, Systemic Inflammatory Response Syndrome; SOFA, Sequential Organ Failure Assessment; ED, Emergency Department; DNA-CPR, do not attempt cardio-pulmonary resuscitation; COPD, Chronic Obstructive Pulmonary Disease.

**Table 2 jcm-08-01337-t002:** Organisms identified by microbiological investigations.

Blood Culture (n)	Sputum Culture (n)	Urine Culture (n)	Wound Culture (n)	Other Culture (n)
*Escherichia coli* (17)	*Respiratory flora* (33)	*Escherichia coli* (33)	Mixed growth (27)	MRSA (4)
Gram positive cocci (16)	*Haemophilus influenzae* (11)	Mixed growth (18)	*Staphylococcus aureus* (16)	*Aspergillus* (2)
*Coagulase negative Staphylococcus* (6)	*Candida albicans* (9)	KESC group (3)	*Coliform* (2)	*Escherichia coli* (2)
MSSA (6)	*Pseudomonas aeruginosa* (9)	*Contamination* (2)	*Serratia marcescens* (2)	*Rhinovirus* (2)
*Pseudomonas aeruginosa* (6)	Mixed growth (8)	*Enterococcus bacteria* (2)	*Skin flora* (2)	*Aggregatibacter aphrophilus* (1)
*Streptococcus pneumoniae* (5)	*Moraxella catarrhalis* (3)	*Pseudomonas aeruginosa* (2)	*Candida albicans* (1)	*Clostridium difficile* (1)
*Klebsiella pneumoniae* (3)	*Staphylococcus aureus* (3)	*Proteus mirabilis* (1)	*Enterococcus* (1)	*Enterovirus* (1)
Gram negative bacilli (2)	*Streptococcus pneumoniae* (2)	*Yeast* (1)	*Escherichia coli* (1)	*Staphylococcus aureus* (1)
MRSA (2)	*Corynebacterium striatum* (1)	No growth (259)	*Group B Streptococcus* (1)	*Streptococcus intermedius* (1)
*Proteus mirabilis* (2)	*Escherichia coli* (1)		*Pseudomonas aeruginosa* (1)	*Yeast* (1)
*Staphylococcus aureus* (2)	*Klebsiella pneumoniae* (1)		*Streptococcus viridans* (1)	No growth (175)
*Enterobacter cloacae* (1)	*Coliform* (1)		No growth (30)	
*Erysipelothrix rhusiopathiae* (1)	No growth (26)			
*Parabacteroides distasonis* (1)				
*Salmonella* (1)				
*Serratia marcescens* (1)				
*Staphylococcus epidermidis* (1)				
*Staphylococcus lugdunensis* (1)				
*Streptococcus agalactiae* (1)				
*Streptococcus intermedius (1)*				
*Streptococcus oralis (1)*				
No growth (344)				

The organisms are listed from the most to least frequently detected by each culture type.

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
