# Peer review of "Real World Patterns of Antimicrobial Use and Microbiology Investigations in Patients with Sepsis outside the Critical Care Unit: Secondary Analysis of Three Nation-Wide Point Prevalence Studies"

_jcm, 2019, doi:10.3390/jcm8091337_

Round 1

Reviewer 1 Report

Minor Comments:

1) Abstract: Page 1, ll. 38: "...antibiotic prescription practice is not guided by microbiological investigations." What do the Authors expect by guidance? It takes 24-48 hours until the results for blood culture are available. Moreover, as demonstrated by the Authors, only few BC were positive. Were all AB prescriptions in negative BCs unnecessary? Clearly not! Hence, the phrasing should be adapted. It is naïve to think that current practice of microbiological investigations are able to guide fast AB therapy of sepsis. However, all AB treatment in septic patients must be accompanied/paralleled by BC tests. That's something different!

2) Study design: General wards and EDs were included. However, the difference between both are not part of the analysis. Please add results for subgroups.

3) Discussion: Page 8, ll. 240-241: "...a firm diagnosis of infection..." What is meant by this phrase? Clinically, microbiologically?? Waiting for BC results is definitely not recommended (see first comment). Please rephrase!

Author Response

Q1: Abstract: Page 1, ll. 38: "...antibiotic prescription practice is not guided by microbiological investigations." What do the Authors expect by guidance? It takes 24-48 hours until the results for blood culture are available. Moreover, as demonstrated by the Authors, only few BC were positive. Were all AB prescriptions in negative BCs unnecessary? Clearly not! Hence, the phrasing should be adapted. It is naïve to think that current practice of microbiological investigations are able to guide fast AB therapy of sepsis. However, all AB treatment in septic patients must be accompanied/paralleled by BC tests. That's something different!

A: Thanks for the thoughtful comments, we agree with the reviewer that we should have phrased this sentence better. We have changed it accordingly.

Q2: Study design: General wards and EDs were included. However, the difference between both are not part of the analysis. Please add results for subgroups.

A: We felt at the preparation of the manuscript, that based on our previous analysis of the same dataset, there is not much difference between ED and ward patients. However as the reviewer specifically requested this, we have prepared a Supplementary Table 1 in the Supplementary appendix to demonstrate the results.

Q3: Discussion: Page 8, ll. 240-241: "...a firm diagnosis of infection..." What is meant by this phrase? Clinically, microbiologically?? Waiting for BC results is definitely not recommended (see first comment). Please rephrase!

A: Thanks for the comment, we agree and it has been corrected. 

Reviewer 2 Report

This manuscript mentioned the indiscriminate use of antibiotics and omission of microbiological investigations in management of septic patients. 

The manuscript points out that microbial testing is not performed properly in patients with suspected sepsis. However, diagnosis by the criteria for sepsis could be unreliable. Antimicrobial agents are also administered when SEPSIS-3 criteria are not met, but it is necessary to judge based on the final infection diagnosis whether the use of these antimicrobial agents is inappropriate. 

Additionally, broad-spectrum antibiotics such as Tazocin are used for initial treatment, while the median duration of use of antibiotics for initial treatment is as short as one day. There are few patients undergoing microbiological testing. Therefore, it is doubtful whether it reflected the population of sepsis appropriately.it will be better to describe the final infectious diagnosis. 

In this study, patients are followed up to 90 days.It is necessary to describe whether there is a difference in the patient's outcome depending on the presence or absence of the antibiotic treatment.

Major comments

Materials and methods part:

It is better to state the reason why patients with mental illness were excluded.

Results part and Discussion part

It is better to include name of diagnosis for infectious disease to assess proper antimicrobial use. 

Table1:

It is better to indicate not range but IQR in age, number of comorbidities, avariety scores.

If possible, it is desirable to include an immunocompromised background in demographics 

because immunodeficiency is a risk of sepsis 

[Page 3, Line 113-180]

If possible, it is desirable to include criteria or indication to start using antimicrobials  

Table2

The state of detection of antimicrobial resistant bacteria (ex ESBL, AmpC, VRE, resistant rate of quinolone in E.coli)should be added in microbiological investigations, because this impormation is important to select type antimicrobial agents.

[Page 7, Line 177-180]

Although the use of the Sepsis criteria is related to the nonuse of antibiotics, whether or not the outcome in each group was influenced by the nonuse of antibiotics to evaluate the effect of antibiotics. Moreover, the difference between the groups with and without antibiotics was not shown, and it is questionable whether each group is similar for their demographics.

Figure1:

In figure, does the ordinate indicate the patients numbers?

It needs to add units to the ordinate in figure.

Conclusion part:

As for SOFA and SIRS, OR is not high although they should be used as a clinical index. it may be not eligible to say that the calculation of SOFA and SIRS contributes to the improvement of the rate of use of antibiotics in patients suspected of sepsis.

Minor comments

Results part:

[Page 7, Line 178−180]

There is a column missing a semicolon in the statistical results. 

Author Response

Q: It is better to state the reason why patients with mental illness were excluded.

A: Thanks for the observation, this has been added to the text.

Q: It is better to include name of diagnosis for infectious disease to assess proper antimicrobial use. 

A: in the majority of the cases the reason to start antimicrobials was “suspicion of infection”, which is as vague as it gets in routine clinical practice. During the data collection, we have attempted to clarify the source of the suspected infection and we present these results in our Supplementary Table 2 in the appendix as requested by the reviewer.

Q: It is better to indicate not range but IQR in age, number of comorbidities, a variety scores.

A: Thanks for the comment, we agree and it has been corrected. 

Q: If possible, it is desirable to include an immunocompromised background in demographics because immunodeficiency is a risk of sepsis 

A: We agree with the reviewer that immunocompromised patients are at higher risk. However we do not have this data readily available. Given that most of our patients were recruited from the normal wards, it is unlikely that we have a large number of these patients in the database.

Q: [Page 3, Line 113-180]

If possible, it is desirable to include criteria or indication to start using antimicrobials  

A: please find our answer to this question above.

Q: The state of detection of antimicrobial resistant bacteria (ex ESBL, AmpC, VRE, resistant rate of quinolone in E.coli)should be added in microbiological investigations, because this information is important to select type antimicrobial agents.

A: Thank you for the question. Probably due to the patient population studied, we have not find any multi-resistant organisms in any of the microbiological cultures. As multi-resistance is relatively rare in the NHS setting we are certain that this general ward population was not readily affected by this.

Q: Although the use of the Sepsis criteria is related to the nonuse of antibiotics, whether or not the outcome in each group was influenced by the nonuse of antibiotics to evaluate the effect of antibiotics. Moreover, the difference between the groups with and without antibiotics was not shown, and it is questionable whether each group is similar for their demographics.

A: Thanks very much for the thoughtful comment. Our regression analysis was confined to understand what affects the antimicrobial administration and we have not attempted to quantify whether antimicrobial management has had any effect on the patient outcome. We feel we have not enough granularity in the data to evaluate this, moreover our previous analysis on the same sample indicates that co-morbidities and pre-admission factors are more likely to have a significant interaction. (Kopczynska BMC Res Notes)

Q: Figure1:

In figure, does the ordinate indicate the patients numbers?

It needs to add units to the ordinate in figure.

A: This has now been corrected.

Q: As for SOFA and SIRS, OR is not high although they should be used as a clinical index. it may be not eligible to say that the calculation of SOFA and SIRS contributes to the improvement of the rate of use of antibiotics in patients suspected of sepsis.

A: Thanks for the comments. We agree that the ORs are not very high for the clinical scores, however we feel that any improvement from the current landscape should be welcomed and the use of these clinical criteria (especially SOFA) could present a “low hanging fruit” to help more discriminate antibiotics use and more thorough microbiological sampling. However we realise that this could be worded better, hence we have changed our conclusion as suggested.

Q: There is a column missing a semicolon in the statistical results. 

A: Thanks for the careful examination of the text, we have added the missing punctuation.

Reviewer 3 Report

Great initiative and analysis

cultures are not indicated in all infection patients receiving antibiotics. it may seem that in the sepsis-3 cohort practice was significantly better 

The broad spectrum AB use in this hospital is concerning and the medical confusion around when to use them is even more intriguing comments have been attached tot he paper and the discussion could be tightened around infection and suspected sepsis cohorts. Are the authors suggesting suspected infection + NEWS>3 = suspected sepsis?

Minor tweaking could get this paper over the line for publication

Author Response

We thank the reviewer for the positive and encouraging feedback. 

We would like to profoundly apologise, as the manuscript submission system did not allow us to learn the specific comments the Reviewer made in the manuscript. We have requested this annotated text from the Journal, however we did not receive it before the deadline of resubmission.

Nevertheless, we have significantly changed the Discussion and removed paragraphs which were repetitive. We hope that the revised version will meet the Reviewer’s approval. Obviously, we would be more than happy to attempt to address any specific concerns, if the Reviewer could kindly provide them via the online system.

To answer the question whether we propose NEWS 3 or above + suspicion of infection = suspicion of sepsis?

Based on our previous work and other recent data on the field, we believe that NEWS 3 + suspicion of infection should trigger the use of Sepsis Screening Tool in all patients. Most probably, in a large proportion of cases, this will NOT lead to diagnosis of sepsis, as there is no organ dysfunction present. However, it appears that using the sepsis screening tool helps to channel the activity towards a more comprehensive review of the patient and more complete diagnostic workup.

Round 2

Reviewer 1 Report

From my point of view, there are no further corrections required

Author Response

Thanks very much for the comments. We are pleased that the Reviewer found that the updated manuscript addressed all the comments raised.

Reviewer 2 Report

In Table 1,It is better to indicate not range but IQR in age, number of comorbidities, a variety scores. It is better to  write  (Q1-Q3) not range.

Author Response

Thanks for the comment, we agree and the table has been corrected.